# Sustainable Hues: Exploring the Molecular Palette of Biowaste Dyes through LC-MS Metabolomics

**DOI:** 10.3390/molecules26216645

**Published:** 2021-11-02

**Authors:** Ralph John Emerson J. Molino, Klidel Fae B. Rellin, Ricky B. Nellas, Hiyas A. Junio

**Affiliations:** 1Secondary Metabolites Profiling Laboratory (SMPL), Institute of Chemistry, College of Science, University of the Philippines, Diliman, Quezon City 1101, Philippines; rjmolino@up.edu.ph (R.J.E.J.M.); kbrellin@up.edu.ph (K.F.B.R.); 2Virtual Biochemical Simulations Laboratory (Good VIBEs), Institute of Chemistry, College of Science, University of the Philippines, Diliman, Quezon City 1101, Philippines; rbnellas@up.edu.ph

**Keywords:** natural colorants, ultrahigh-performance liquid chromatography–high resolution mass spectrometry (UHPLC–HRMS), metabolomics, GNPS, molecular networking

## Abstract

Underutilized biowaste materials are investigated for their potential as sustainable textile colorants through an approach based on mass spectrometry, bioinformatics, and chemometrics. In this study, colorful decoctions were prepared from the outer bark of *Eucalyptus deglupta* and fruit peels of *Syzygium samarangense, Syzygium malaccense, Diospyros discolor*, and *Dillenia philippinensis.* Textile dyeing was performed along with liquid chromatography–mass spectrometry (LC–MS)-based untargeted metabolomics to determine the small molecules responsible for the observed colors. Global Natural Products Social Molecular Networking (GNPS) guided the annotation of black-producing proanthocyanidins in *D. philippinensis* and *E. deglupta* through complexation with FeSO_4_ mordant. Flavonoids from the yellow-colored *D. philippinensis* extracts were found to be similar to those in *Terminalia catappa*, a known traditional dye source. A higher intensity of epicatechin in *E. deglupta* produced a red-brown color in the presence of Cu^2+^. Furthermore, *Syzygium* fruit peels have poor wash-fastness in cotton fibers, but bioactive chalcone unique to *S. samarangense* samples may be a potential nutritional food colorant. Unsupervised PCA and supervised OPLS-DA chemometrics distinguished chemical features that affect dyeing properties beyond the observed color. These findings, along with growing data on natural dyes, could guide future research on sustainable colorants.

## 1. Introduction

Sustainability with sociocultural relevance reinvigorated research and development of natural dyes and pigments [1,2]. Additionally, with the detrimental effects on the environment and human health associated with synthetic dyes, there is a renewed interest in the discovery of natural biocolorants [1,2]. Guiding the discovery process is a wealth of literature on plant [3], fungi, and bacteria [4] and animal sources [5,6]. Exciting applications are being explored for leather tanning [7], food, and cosmetics [8,9], as well as in sensor technology [10] and the development of dye-sensitized solar cells [11].

In the Philippines, the promotion of natural dyes supports the livelihood of indigenous groups and highlights cultural heritage through their woven textiles [12]. Furthermore, the practice of traditional dyeing serves as a platform for promoting cultural heritage [12]. In line with their importance, the Philippine Textile Research Institute (PTRI) published a compendium detailing traditional and scientific knowledge on over one hundred dye-yielding plants [13,14]. To further promote sustainable dyeing, the existing list could be appended with colorants sourced from waste materials from food and timber production [1]. Plant biowastes had very little to no documented applications, such as peels from native fruits *Syzygium samarangense* (Blume) Merr. and Perry. (loc. makopa), *Syzygium malaccense* (L.) Merr. and Perry. (loc. makopang-kalabaw), *Diospyros discolor* Willd. (loc. mabolo), and the endemic *Dillenia philippinensis* Rolfe (lockatmon). Similarly, there is a need to explore the use of the outer bark of *Eucalyptus deglupta* Blume (Mindanao rainbow tree), utilized commonly for its pulp and timber [15].

Perceived color and application of natural dyes to textiles depend heavily on their chemical composition [16]. In terms of plant-derived colorants, specific classes of compounds (flavonoids, tannins, carotenoids, etc.) are known to impart color to fibers in the presence or absence of mordants [4,16]. Additionally, present in some dye baths are other small molecules such as hydrophobic lipids that could affect the interaction of the chromophores on fibers [17]. Untargeted small molecule analysis, therefore, is crucial to understand the complex color yielded by crude natural dyes [18]. LC-MS-based metabolomics presents a sophisticated platform for the sensitive detection, characterization, and comparison of small molecules present on natural dye samples [18]. Through state-of-the-art instrumentation in the form of ultra-high performance liquid chromatography in tandem with high-resolution mass spectrometry (UHPLC–HRMS), the mass spectra of small molecules can be acquired without extensive purification [19]. Data analysis using the cloud-based Global Natural Products Social Molecular Networking (GNPS) platform allows accessible and free matching to a community-curated spectral library for putative identification of metabolites [20]. In addition, molecular networking analysis indicates structural relationships of annotated metabolites with unidentified constituents that may be present on different dye samples [20]. The distribution of specific metabolites across groups of samples could also be further explored using multivariate statistics [21]. This study integrates LC-MS profiling, molecular networking analysis, and chemometrics to annotate the color palette yielded by biowaste dyes, and the strength of the metabolomics approach to compare spectral signatures across diverse samples and identify significant features that could be linked to similar or divergent dyeing properties. Because of these, newly identified biocolorants from *S. malaccense*, *S. samarangense*, *D. discolor*, *D. philippinensis*, and *E. deglupta* were not only analyzed side-by-side with each other, but also compared with the molecular fingerprint of a traditional source of yellow dye, *Terminalia catappa* L. (talisay).

## 2. Results

### 2.1. Extraction of Biowaste Natural Dyes and Dyeing on Textiles

The main driving force towards exploration of biowaste natural dyes is the desire to provide sustainable alternatives to synthetic dyes as well as plant-based sources that require months or even years of cultivation [4]. Furthermore, utilizing food wastes could result in profit among small- to large-scale enterprises even with limited investments in terms of capital [8,22]. In this research, we explore underutilized sources from fruit peels of *S. malaccense*, *S. samarangense*, *D. discolor*, *D. philippinensis,* and the outer bark of *E. deglupta* (Figure 1) that are easily accessible to different locals throughout the Philippines [23,24]. The highlighted extraction process was the decoction preparation based on the traditional method of many communities. The mentioned approach is relatively cheap, safe, and readily accessible compared with the use of organic of solvents [25].

Heated aqueous extraction was able to produce colorful solutions that changed colors in the presence of FeSO_4_, alum (KAl(SO_4_)_2_·12H_2_O), CuSO_4_, tartaric acid (C_4_H_6_O_6_), and soda ash (Na_2_CO_3_) mordants (Figure 2A). These fixatives were used in the pre-treatment of fabrics (pre-mordanting) and were also incorporated during dyeing process (meta-mordanting). As can be seen in Figure 3, the addition of mordants increased the interaction of the extracts with cellulose fiber and resulted in a diverse range of color. Exceptions are *Syzygium* fruit peels, whose color readily washes away from the fabric after the dyeing process. Observed colors are further described elsewhere (Appendix A) in terms of their hue, saturation, and value coordinates. Significant coloration and overlap were observed among *E. deglupta*, *D. philippinensis*, and *T. catappa*. Annotation of chemical features from the biowaste extracts became the focus of the succeeding metabolomics investigation.

### 2.2. UPLC^®^-ESI qTOF Profiling, Molecular Networking and Structural Analysis of Small Molecules from Extracted Natural Dyes

Similarities and contrast in the color schemes (Figure 3) motivated the metabolomics investigation of the samples. Reverse-phase LC-MS profiling showed the difference in the small molecule fingerprint of the biowaste dyes (Figure 4). Annotation of relatively abundant features was carried out through library search in GNPS. A wide range of compounds was putatively identified, including highly polar anthocyanidins, anthocyanins, sugars, amino acids, and other primary metabolites. In addition, diverse structures of flavonoids and tannin dyes were noted for most of the samples at t_R_ = 3.00–5.00 min. Present only in *S. samarangense* are low molecular weight chalcones (t_R_ = 3.00–5.00 min, Figure 4F) immediately followed by highly retained lipids, and chlorophyll pigments and their catabolites (Figure 4A–F, t_R_ = 7.50–9.00 min). Details supporting the annotation of these features are tabulated in Appendix A and Appendix A. 

Structural relationships among annotated and unidentified features were further explored through molecular networking analysis in GNPS [20]. In a molecular network such as in Figure 5A,B, precursor ions are represented as nodes, which are connected if spectral similarity among them exceeds a set threshold [20]. A cut-off value of 0.70 was used in the analysis, as suggested from previous reports. [26,27]. The cosine score calculated from experimental data is specified in the thickness of the edges connecting the nodes [26]. Since the fragmentation pattern arises from the molecular structure, spectral families hint at precursor ions that share similar backbone or functional groups.

The molecular network in Figure 5A presents the occurrence of phenolic dye components such as anthocyanins and flavonols present in the samples. Annotated features (labelled accordingly) are associated with mirror matches such as in Figure 5C,D for cyanidin (t_R_ = 1.05 min., 0.31 ppm, cos score = 0.77), and cyanidin-3,5-O-diglucoside (t_R_ = 1.05 min, 0.23 ppm, cos score = 0.87). Clustering of cyanidin with flavonols (quercetin, kaempferol, myricetin, and isorhamnetin) is expected due to similarities in molecular structure, giving rise to shared identical product ions formed from fragmentation routes such as retro-Diels Alder (RDA) and decarbonylation reaction [28]. Networking of glycosylated flavonol, anthocyanins, and their aglycones is attributed to intense product ion signals from the aglycone moiety as well as prominent neutral losses of 146, 162, and 178 Da from the elimination of rhamnose, glucose, and glucuronic acid along the glycosidic bonds, respectively [29]. Isobaric features with *m/z* 383.0600 (t_R_ = 8.41 min. and 9.07 min) as well as *m/z* 367.0118 (t_R_ = 8.87 min.) have no library hits but were annotated based on their networking with both quercetin and kaempferol (Figure 5A). Neutral loss of 80 Da between precursor ion and *m/z* 303.0446 hinted at a sulfate modification on one of the hydroxyl groups of quercetin [30]. Location of substitution is deduced using the prominent signal of *m/z* 301.0334 present only on this isomer (Figure 5F). Figure 5E explains the formation of this product ion from the simultaneous loss of SO_3_ and H_2_ of the precursor ion. Fusion of ring systems B and C of the flavonoid is highly likely if the sulfonation site is at C-3. Therefore, annotations for *m/z* 303.0446 and 367.0118 are quercetin-3-O-sulfate (1.93 ppm) and kaempferol-3-O-sulfate (0.90 ppm), respectively. 

Analysis of phenolic dyes in Figure 5A were then extended to other precursor ions that include a sugar group in their fragmentation pattern. For *D. discolor*, a unique precursor ion (Appendix A) with *m/z* 355.0651 fragmented to produce *m/z* 203.0508 associated with sodiated hexose, [Hex + Na]^+^. In addition, the neutral loss of 152 Da from both *m/z* 355.0651 and *m/z* 345.0215 precursor ions indicated gallate substitution [31]. Because of this, annotation provided for the former is glucogallin (2.76 ppm), whereas the latter, which exhibited *m/z* 153.0192 (C_7_H_5_O^+^, Appendix A), is characterized to be digallic acid (2.90 ppm). Annotation of these low molecular weight derivatives of gallic acid is important as they could impart black coloration in the presence of FeSO_4_ [32]. 

Compounds with high molecular weight showed distinct clusters with molecular networking. Spectral families in Figure 6C,D include precursor ions common between *T. catappa* and *D. philippinensis*, which can be implicated in the comparable color profiles resulting from the two samples. GNPS annotated vitexin (Figure 6C) in *D*. *philippinensis*, whereas isovitexin (Figure 6D) gave high signal intensity in *T. catappa*. These isomers yielded identical product ions characteristic of C-glycosylated flavonoids [33]. Vitexin is differentiated by a high intensity of dehydration reaction products from C-8 glycosylated flavonoids [33,34]. Cross-ring cleavage of the pyran ring, on the other hand, is favored for C-6 glycosylated flavonoids as can be seen for isovitexin (Figure 6D) [33,34]. This difference in product ion intensities was used to characterize the glycosylation sites on the other analogs (Figure 6A). Unmatched precursor ions with *m/z* 579.1496 displayed a similar fragmentation pattern as vitexin (Figure 6E). Neutral loss of 146 Da and the presence of *m/z* 147.0436 were linked to the presence of a rhamnose group (Figure 6F) [29]. The location of this modification was also identified to be C-2 of the pyran ring, based on one of the cross-ring cleavage products with *m/z* 459.1169 (Figure 6F) [33]. Hence, the annotation given to *m/z* 579.1496 is vitexin-2”O-rhamnoside (1.55 ppm). A similar approach was used to characterize *m/z* 585.1265 and *m/z* 601.1261, which resembles the fragmentation pattern of vitexin and isoorientin, respectively. Again, neutral loss of 152 Da, and intense *m/z* 153.0192 product ion resulted in the annotation of the precursor ions as vitexin-2”O-gallate (*m/z* 585.1265, 3.59 ppm) and isoorientin-2”O-gallate (*m/z* 601.1261, 3.66 ppm). Vitexin-O-sulfate was also characterized based on its network connection with vitexin, and neutral loss of 80 Da. Position of the sulfate group cannot be determined due to the absence of diagnostic productions. 

Figure 6B, on the other hand, showed the presence of additional high molecular weight tannins that are unique to *T. catappa*. Presence of *m/z* 303.0161 (Appendix A) hinted to an ellagic acid core that served as the basis for the structural characterization of terminalin (*m/z* 603.0092, 0.61 ppm, Appendix A), a known ellagitannin from *Terminalia* sp. [35,36]. Using the MS/MS profile of terminalin, two other nodes in the cluster were putatively assigned as terminalin-O-glucose-O-hexahydrodiphenic acid (*m/z* 1066.0640, 1.27 ppm) and terminalin-O-glucoside (*m/z* 765.0583, 173 ppm) based on the subsequent neutral loss of hexahydroxydiphenic acid (302 Da) and a glucose (162 Da) unit (Appendix A) [31]. These two compounds have not been previously reported in *T. catappa*.

Meanwhile, the tannin profile of *D. philippinensis* fruit peel is comparable to *E. deglupta* bark. The network in Figure 7A contains *m/z* 579.1483 (Figure 7B), which matches with procyanidin B2 on the GNPS library. Remaining nodes that were unidentified were annotated as proanthocyanidins, consisting of epicatechin, epigallocatechin, and epiafzelechin subunits based on neutral losses and product ions derived from quinone-methide (QM) fragmentation [36]. Additional reactions that supported the structural analysis of these phenolic compounds include RDA, benzofuran forming fusion (BFF), and heterolytic ring fusion (HRF) reactions [36] (Appendix A). Structural analysis of *m/z* 731.1584 is shown in Figure 7C. Presence of epicatechin subunits is supported by the detection of *m/z* 289.0714 product ion derived from the QM cleavage of the non-hydrolyzable backbone. In addition, the presence of gallate substitution was inferred from neutral loss of 152 Da forming *m/z* 579.1154. The exact position was determined based on the higher relative intensities of the RDA products (*m/z* 427.1002 and *m/z* 409.0907) compared with *m/z* 519.1154 (loss of galloyl). Due to extensive hyperconjugation of the resulting product, it is well established that RDA reaction is favored over gallate elimination if the top-unit is galloylated [37,38]. Annotation for *m/z* 731.158, therefore is epicatechin-3-O-gallate epicatechin (3.09 ppm), which resembles the structure of complex proanthocyanidins common to *T. catappa* and *D. philippinensis* (Figure 7). 

Interestingly, small molecular weight flavan-3-ols that are precursors to complex tannins [36] were found to be significantly abundant in *E. deglupta* (Figure 7D). This observation is important because catechin and epicatechin are known to give red-brown coloration to textiles in the presence of CuSO_4_ mordant [39]. This shade was observed only on cotton fiber dyed with *E. deglupta* extract (Figure 3). Annotation of similar and unique features among dye samples prompted further examination of their chemical profile in terms of multivariate statistics.

### 2.3. Chemometrics Analysis of Natural Dye Samples

Unsupervised multigroup analysis in XCMS Online unraveled the molecular relationships between various biowaste colorant sources. Non-parametric principal component analysis (PCA) showed that the first two principal components (PCs) only accounted for 45.37% of the total variance in the chemical profiles, prompting the consideration of PC 3 (16.45%; Figure 8A). Centering and log transformation was also carried out to account for the heteroscedasticity (i.e., inconsistent variance across concentration levels) of LC-MS data [40]. Putatively identified compounds from the GNPS molecular networking analysis such as flavonoids served as basis of comparison of these biocolorant sources. Various intensities of these compounds could likely influence the perceived color of the extract with or without mordant.

Plant extracts spread in three distinct regions (Figure 8A): R1 (PC1: −25.0 to −5.0, PC2: 5.0 to −2.5, PC3: 15.0 to −20.0), R2 (PC1: 0.0 to 15.0, PC2: 7.5 to 25.0, PC3: 0.0 to 5.0), and R3 (PC1: 0.0 to 10.0, PC2: 0.0 to −20.0, PC3: 15.0 to −20.0). Major variance contributors are phenolics, particularly flavonoids and their glycosylated derivatives. These compounds along with lipid analogs and several other unidentified metabolites falling within the *m/z* 200 to 800 mass range contribute greater than or equal to 1.00% of the total variance observed in PC1 and PC2 (Figure 8B, data points in red). An example of lipid analogs identified by XCMS Online and molecular networking is dilinolein (*m/z* 613.4847, *p*-value: 1.92 × 10^−8^), which is found only in *T. catappa* and *D. philippinensis* extracts. Additionally, the same set of compounds account for most of the variance in PC3 (data not shown). Although these compounds greatly contribute to the spread of data in PCA, they do not directly cause the observed color of the extracts but may influence solubility. For lipids, further studies are necessary to verify their effects on the dye properties.

Leaf extracts from *T. catappa* and *D. philippinensis* cluster in R1, with the latter exhibiting a bimodal distribution (Figure 8A). Clustering in R1 is expected from leaf samples due to the presence of chlorophyll pigments and their catabolites in the form of pheophytin and pheophorbide. Moreover, the effect of phenolics in R1 can only be observed when PC3 is considered (Appendix A). It is interesting to note that *D. discolor* fruit peels (R2) did not cluster with *Syzygium* fruit peels (R3) in Figure 8A despite physical similarities, highlighting the effect of the presence and abundance of key flavonoids in group behavior. The abundance of known colored compounds such as flavonoids in certain extracts could also influence the observed color. However, PCA cannot predict the resulting color upon reaction with a mordant during textile applications.

Box-and-whisker plots (Appendix A) also show relative abundances of putatively identified phenolics that have known contributions to dye production. Table 1 summarizes these chemical features that contribute significantly to the behavior of samples observed in PCA. Vitexin/isovitexin is highly abundant in *T. catappa* leaf and *D. philippinensis* fruit peel extracts. Meanwhile, a compound annotated by GNPS to be unique to *S. samarangense* fruit peel is 2,4,6-trimethoxychalcone, a flavonoid analogue that has a known yellow color [41]. However, this compound is highly hydrophilic and does not interact well with the fabric and the mordant. Another compound that is also abundant in *S. samarangense* fruit peel is quercetin. Despite the ubiquity of this compound in terrestrial natural products [42], it contributes 4.25% in total variance, and is found in relatively low abundance in *D. discolor* fruit peel compared with *S. samarangense*. Similarly, its glycosylated derivative, quercetin-O-arabinoside is highly abundant in *S. samarangense* but absent in *D. discolor* and *D. philippinensis.* These compounds are also highly hydrophilic and may have contributed to the low intensity or absence of yellow coloring when *S. samarangense* is reacted with alum. Tiliroside, a glycosylated flavone, is detected only in the fruit peel of *D. philippinensis* and does not have any reported use as a textile colorant.

Furthermore, the presence of certain metabolites and their effect on the produced coloration were highlighted by orthogonal projection to latent structures discriminant analysis (OPLS-DA). Extracts that produce black coloration upon addition of FeSO_4_ mordant distinctively separate from samples that do not (Figure 8C). Polyphenols with masses *m/z* 867.2146, 579.1483, and 585.1257 were putatively identified as epicatechin, procyanidin B2, vitexin-2”-O-gallate (Figure 8D), respectively. These compounds are associated with the production of black coloration in complexation reactions with iron mordants. Flavonoid isoorientin is present in *T. catappa* and *D. philippinensis*, both of which are producing black coloration with FeSO_4_ mordant. However, it is found to be absent in *E. deglupta*, another black-producing extract. This suggests that isoorientin, despite its relative abundance in other black-producing biowaste sources, is not a contributing feature in OPLS-DA.

## 3. Discussion

Chemistry of the colors of natural dyes is due to the presence of light-absorbing molecules in the crude extracts [4]. These organic compounds have chromophore motifs with alternating single and double bonds [43]. Extensive conjugation decreases the energy needed to promote electrons from the highest occupied molecular orbital (HOMO) to the lowest unoccupied molecular orbital (LUMO), thereby resulting in absorbance of wavelengths within the visible range of the spectrum [43]. In addition, the presence of auxiliary groups (-OH, -NH_2_, -CH_3_) also affects the wavelength of energy absorbed by natural dyes [43]. Detailed studies explored the effect of solvent, pH, and mordants on the color and interaction of natural dyes in solution and/or in textiles [44,45,46,47,48]. This existing knowledge on the structure and physical chemistry of plant colorants guided the screening of biowaste extracts as natural dyes through metabolomics.

Identification of pigment classes such as flavonoids, tannins, and chalcones accounted for the colors obtained from the dyestuffs. These compounds were easily extracted as decoctions, with hydrogen bonding interactions with polyphenolic compounds [4]. Aside from its role in extraction, the chosen solvent significantly influenced color such as the strong red obtained from *Syzygium* fruit peel [44,48]. In silico simulations of dyes in the gas phase predicted blue color in vacuum that is shifted to red wavelengths in the presence of explicit water and methanol solvent [44]. The importance of solvent lies in its role in increasing the polarizability of dye molecules and incorporating thermal fluctuations and conformational variability in the molecules, resulting in the experimental observations [45,49]. Considering this, the resulting spectra of anthocyanins have been correctly predicted with explicit models of water or methanol solvent [44]. In addition, solution pH can slightly, or drastically, change the structure of the chromophore [4]. For cyanidin, saturation of red color is most intense at highly acidic conditions (pH 3.0), where the flavylium cation of the anthocyanidin dominates [49]. In the presence of alkaline Na_2_CO_3_, the flavylium ion is converted to an unstable hemiketal, which further reacts to yield a chalcone [49]. Equilibrium among these different species explained the green dye bath from *S. malaccense* pre-mordanted with Na_2_CO_3_ (Figure 3). In contrast, yellow green color was observed in *S. samarangense* due to the inherent presence of yellow chalcones not present in *S. malaccense* (Figure 3).

Even with pre- and meta-mordanting, extracts from *S. samarangense* and *S. malaccensse* did not impart significant coloration to textiles (Figure 3). Limitation on the use of the *Syzygium* samples can be attributed to the higher solubility of anthocyanins in the dispersion medium compared with the cellulose fibers [50,51]. A previous report circumnavigated this using SnCl_2_ at acidic conditions [52]. This mordant, however, is rarely used by Filipino artisans, and therefore was not explored in the study. Interestingly, the presence of chalcones did not impart significant yellow coloration to cotton dyed with *S. samarangense* extract [41]. Chalcones gave the most intense signals in the base peak chromatogram of the samples (Figure 4F), although signal intensities do not necessarily correlate with the abundances of these compounds in the crude mixture. It is also possible that the presence of anthocyanins reduces the dyeing capabilities of chalcones through the formation of colloidal dispersion [53], although this aspect is no longer explored in this work.

On the other hand, the presence of flavonoids (quercetin, vitexin, their glucosides) imparted yellow coloration for both *D. philippinensis* and *T. catappa*. The presence of these flavonoids was corroborated by the UV-Vis absorbance profile showing maximum absorption wavelengths at 275 and 373 nm corresponding to the cinnamoyl group (rings A and C) and the benzoyl functional group (rings B and C) of the flavonoids, respectively [54] (Appendix A). In terms of solvent effects, polar protic solvents stabilize flavonoids through H-bonding interactions, resulting in a slight shift in the absorbance towards longer wavelengths [55]. A more pronounced red shift emerges in the presence of mordants, which stabilizes flavonoids through metal-to-ligand charge transfer [56,57,58,59,60,61,62,63]. Hyperchromic shift was also observed on the dyebaths (Appendix A) indicating possible coagulation of dye molecules in the presence of mordant [53]. Such aggregation promotes the dyeing process and the mordants follow this up by effectively fixing the dye molecules on the cellulose fiber [51].

Previous reports showed that metal ions (Fe^2+^, Al^3+^, and Cu^2+^) through three hydroxyl-containing sites on the flavonoid structure [56,57,58,59,60,61,62,63]. Complexes with 1:2 stoichiometry between metal ion and flavonoid are formed through chelation in the oxygen atoms of C3-C4 (ring C) or between C4-C5 (ring A-C) [57,58,59,60,61,62]. Meanwhile, a stable 1:1 complex is formed through coordination at C3′-C4′ (ring B) [57,58,59,60,61,62]. For both *T. catappa* and *D. philippinensis*, annotated flavonoids existed in the O- and C-glycosylated forms. These derivatives have been previously noted to have lower affinity for mordants, especially if the glycosylation site is a hydroxyl group involved in complexation [63]. Sugar chain/s of glycosylated flavonoids do not contribute to the HOMO and LUMO but enhance nucleophilicity via electron donation, especially if the position of the sugar group is at C7 [63].

Contrary to *T. catappa* and *D. philippinensis*, *E. deglupta* did not produce yellow in the presence of alum mordant (Figure 3). Highly intense features in *E. deglupta* are epicatechin and catechin that lack carbonyl group at C4, which is crucial to chelation with Al^3+^ [56] Sole binding site for metal is the catechol group that could interact with Cu^2+^ ions. This weak catechol-metal complex which becomes oxidized gives red-brown shades observed in textiles due to catechinone [64]. For *T. catappa* and *D. philippinensis*, a local maximum at 293 nm emerged from the oxidation of quercetin that was implicated to the brown color in the presence of excess CuSO_4_ [61].

The complexation process is also responsible for black shades obtained from the decoctions in the presence of FeSO_4_ [65]. The dark color could be from the complexation of FeSO_4_ with either gallic acid or catechin [31,66]. The computational study also highlights the role of water molecules in maintaining stable octahedral geometry across the metal ion during the complexation process [59].

Multivariate analysis using PCA [67] and OPLS-DA [68] was able to verify the effect of known color-producing compounds such as flavonoids and flavotannins. However, factors such as ionization efficiency and concentration may affect the chemical features detected by these statistical methods. For PCA, most of the features that heavily contribute to variance may not directly produce a color nor complex with mordants during the dye process. While powerful, PCA can only reveal differences between samples if the features are major contributors to the total variance [67,68]. OPLS-DA compensates for this by distinguishing the samples based on group-specific characteristics, in this case, dye molecules and their associated color [68]. The distinction between the extracts is their explicit production of black coloration upon complexation with iron mordant. Black coloration was obtained from Fe^2+^-complexed extracts from *E. deglupta*, *D. philippinensis,* and *T. catappa* and their separation from *S. samarangense, S. malaccense,* and *D. discolor* was explained by the abundance of epicatechins, procyanidins B2, and vitexin-2”-O-gallate.

Future research could also look at the effect of non-pigment molecules on the dyeing properties of crude samples. A molecular network in Appendix A contains high molecular weight phosphocholine type lipids detected mostly on *D. philippinensis* fruit peel.

Multiple reports describe beneficial effects of liposome formation on the dyeing process, especially on wool fibers [69,70,71]. The presence of lipids was shown to promote dye interaction with fibers through vesicle formation and release that can be fine-tuned by adjusting pH or dyeing temperature [71]. Adding to this, liposome-assisted dyeing promotes more sustainable energy use since it employs milder dyeing conditions at lower temperatures (usually 10.0 °C lower than conventional processes) [71]. With higher concentration of lipids, however, liposomes exhibit greater stability, and the dyeing properties of the extracts decrease [71]. Succeeding work could use targeted mass spectrometry-based metabolomics towards quantifying and understanding their effect on the color yield of different natural dyes.

Furthermore, the results of the study also highlight potential challenges associated with the use of some of the biowaste dyes. Chief dye molecules annotated were flavonoids known to have poor photostability [57]. To determine lightfastness, additional experiments could investigate the optimum concentrations of metal ion and dye that will minimize photobleaching [72]. Use of ionic [73] and biomordants [74] could also be explored together with the incorporation of UV-quencher compounds such as nickel hydroxyl-arylsulfonate [75].

These suggestions could be included in future research to address the major limitation of this work in terms of assessment of the fastness properties of the samples. Despite this, the major contribution of this metabolomics study is on the preliminary mass spectral data of biowaste natural dyes that will be deposited in GNPS. This additional information on natural dyes could be used by other groups exploring unique and sustainable sources of colorants.

To summarize, LC-MS based metabolomics was able to guide rapid screening of biowaste materials as alternative source of colorants. Comparative analysis in terms of fragmentation pattern and chemometrics also provided insights behind the color produced by the samples. Results of untargeted profiling also revealed the presence of highly intense compounds such as anthocyanins, phosphocholine lipids, and flavonoids that could influence color during and after the dyeing process. Overall, small molecule fingerprinting provided extensive chemical information on natural dyes that could be useful for future research in process development, quality control, and the discovery of sustainable hues.

## 4. Materials and Methods

### 4.1. Chemicals and Reagents

Acetonitrile and water solvents used in sample extraction and chromatography were from Merck (Darmstadt, Germany). Formic acid incorporated in the mobile phase, as well as CuSO_4_·5H_2_O, KAl(SO_4_)_2_·12H_2_O (alum), and FeSO_4_·2H_2_O used in the dyeing experiments were acquired from Thermo Fisher Scientific (Waltham, MA, USA). Meanwhile, standard compounds were sourced from Sigma Aldrich (St. Louis, MO, USA)

### 4.2. Extraction of Natural Dyes

Fruits of *S. malaccense* and *D. discolor* as well as bark of *E. deglupta* were collected from the municipality of Naga, Zamboanga Sibugay. Meanwhile, fruit peels of *D. philippinensis,* and *S. samarangense,* and leaf of *T. catappa* were sampled inside the UP Diliman Campus. Specimens were identified at the Jose Vera Santos Memorial Herbarium, Institute of Biology, UP Diliman.

Natural dyes were then immediately extracted by soaking fresh plant parts in water following the 1:20 (g/mL) ratio. The mixture was heated at 100 °C for 30 min, after which the solutions were filtered and lyophilized. Appropriate amounts of dried extracts were weighed and resuspended in 50:50 acetonitrile: water to obtain a final concentration of 3 mg/mL for LC-MS analysis.

### 4.3. LC-MS Analysis

Instrumental analysis was performed based on a method that has been optimized for natural dyes [18]. Profiling of extracts were carried out using Waters Acquity^®^ Ultrahigh Performance Liquid Chromatography (UPLC^®^) system in tandem with a Xevo G2-XS quadrupole time-of-flight mass spectrometer. Detection of small molecules was performed in the positive mode electrospray ionization (ESI) source. To obtain optimum separation of metabolites, chromatographic separation involved injection of 3.5 µL extracts in a Waters Acquity^®^ CSH Fluorophenyl column (1.7 μm, 50 mm × 2.1 mm) maintained at 30 °C. Compounds eluted through a steady stream (0.35 mL/min) of binary mobile phase system that consisted of acetonitrile and water both infused with 0.1% formic acid. Composition of organic solvent were 5% at 0.00–0.50 min, 20% at 2.00 min, 50% at 4.00–5.50 min, 100% at 7.50–9.00 min, and back again to 5% at 10.00–10.50 min.

Mass spectrometry analysis in the positive ionization mode was carried out with the following tune settings: Capillary voltage of 3.20 V, cone voltage of 45 V, source offset of 80 V, and source and desolvation temperature of 150 °C and 500 °C, respectively. Full-scan analysis was performed in the sensitivity mode of the instrument within *m/z* 50–1500 range and scan time of 0.50 s. Meanwhile, MS/MS analysis was accomplished using the Data-Dependent Acquisition (DDA) mode of the instrument. Selection of precursor ions were guided by a signal intensity threshold of 3.0 × 10^5^ and sampling only the eight most abundant ions per scan. Fragmentation patterns were obtained through collision induced dissociation with argon curtain. To obtain comprehensive information for the different metabolites, profiling runs were executed under different collision energies (15 eV, 15–30 eV, 30–45 eV, and 45–60 eV). The same experimental method used to profile crude natural dye samples was used to analyze standard compounds. 

### 4.4. Bioinformatics Analysis

DDA data in the form of Waters .RAW file were converted to open-source 32-bit .mzXML file format using the MSConvert tool of ProteoWizard MSConvert Version 3 Software [76]. Datasets on natural dyes were then uploaded and subjected to library search and molecular networking analysis in the cloud-based GNPS platform [21]. The library search workflow was used to provide putative identifications for the metabolites. Spectral hits that were displayed were limited to those with precursor and product ion deviation of less than *m/z* 0.02 and *m/z* 0.05, respectively. At least eight matched peaks also existed between sample and reference file, and the cosine (similarity) score exceeded 0.70 [20,27]. On the other hand, molecular networking generates consensus spectra for identical MS/MS scans that are within the same precursor ion and fragment ion mass tolerance. Precursor ions, represented as nodes, were then connected only if they would exceed six matched peaks and a cosine score of 0.70. Maximum number of connections (topK) for each node were set at seven, and the maximum size of a single cluster was limited to 300 nodes [20,27]. After analysis in GNPS, the generated molecular networks were exported in the .graphml format to further customize network visualization on Cytoscape [77].

### 4.5. UV-Vis Spectrophotometric Analysis

Absorbance of extracted dyes of ultraviolet and visible radiation were investigated using a stand-alone Shimadzu Double Beam UV-Vis spectrophotometer. Colorant solutions were prepared in the concentration range of 32.5–250 ppm by way of serial dilution with water (Merck Lichrosolv, Kenilworth, NJ, USA). Samples were loaded in a one (1) cm quartz cuvette and the absorbance measurements were recorded in the 250–800 nm wavelength range.

Succeeding experiments explored the interaction of biowaste colorants with common dyeing mordants. Profiling of unmordanted extracts revealed that a solution concentration of 250 ppm gave highly sensitive signals that do not exceed one absorbance unit. Mordanting with FeSO_4_, alum, and CuSO_4_ involved the mixing of 250 ppm natural dye with varying concentrations (10, 50, 125, 250, and 350 ppm) of mordant. The set-ups could react for 10 min. After centrifugation, the supernatants were subjected to UV-Vis analysis.

### 4.6. Textile Dyeing and Colorimetric Evaluation

Dyeing experiments referred heavily on a reported method for natural dye application using cotton [65]. Total sizes of 4 × 4 cm pieces of cotton fabric were pretreated by immersing in boiling solutions of mordants (5 mg/mL FeSO_4_, alum, CuSO_4_, tartartic acid, and sodium carbonate) for 1 h. After drying, fabrics were submerged in dye baths (1:10 liquor ratio), which consisted of 3 mg/mL extract and 3 mg/mL mordant. Dyeing was carried out for 1 h under a heating mantle that was kept at 100 °C. The dyed pieces of cloth were then dried and washed three times with laboratory detergent. Colorimetric assessment of dyed fibers was carried out by taking scanned images of the dyeing panels and analyzing reflected color using the Digital Color Meter available on MacOS (Apple Inc., Cupertino, CA, USA).

### 4.7. Multivariate Statistics

XCMS Online [21] was used for the multivariate analysis of MS^1^ centroid data. Raw full scan data was converted to 64-bit .mzxml files via ProteoWizard MSConvert Version 3 Software [76]. Feature detection parameters include 5.0 ppm for maximal tolerated *m/z* deviation between consecutive scans, minimum and maximum peak widths of 2.0 s and 15.0 s, respectively, signal-to-noise threshold of 1.0 × 10^2^ prefilter intensity of 1.0 × 10^5^, and noise filter of 1.0 × 10^3^. These parameters were based on defaults suggested by XCMS for Waters high-resolution data [21] and were tweaked slightly to suit the samples for this study. Peak integration and normalization are built into the XCMS workflow. The Kruskal–Wallis non-parametric test was selected with *p*-value thresholds of significant and highly significant features set at 0.05 and 0.005, respectively. Fold-change threshold for highly significant features is 1.5 and greater. Annotation parameters are 5.0 ppm error and 0.015 ppm for *m/z* absolute error. A 100s width was considered for extracted ion chromatograms. PCA scores and loadings plots were log-transformed and centered and were replotted using OriginPro 8.5 (OriginLab Corporation, Northampton, MA, USA).

Supervised analysis was conducted using Metaboanalyst 5.0 [78] (https://www.metaboanalyst.ca, accessed on 15 August 2021). Converted Waters .RAW files were compressed and uploaded to the Metaboanalyst server with .txt metadata on color produced via FeSO_4_ complexation. centWave was chosen as a peak detection method, with 0.01 *m/z* difference, 10.0 signal-to-noise threshold, 10.0 base width, 5.0 ppm error tolerance, 5.0 and 30.0 minimum and maximum peak widths, and 1000.0 noise filter. While the peak detection algorithm of Metaboanalyst is the same with XCMS Online, some parameters from the latter are not compatible with Metaboanalyst. Consequently, parameters used in Metaboanalyst were optimized independently from XCMS Online.

## Figures and Tables

**Figure 1 molecules-26-06645-f001:**
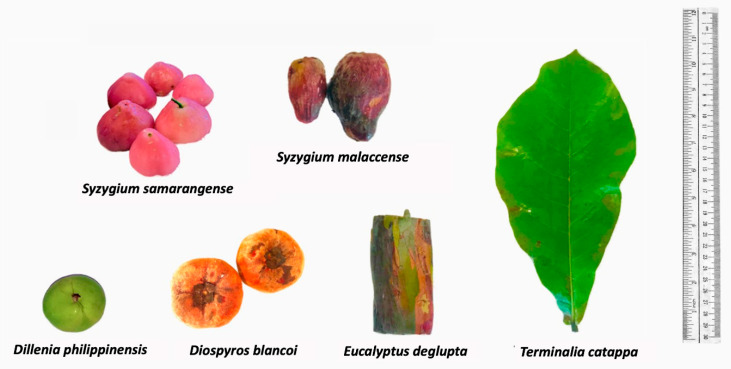
Plant sources of extracted natural dyes. Colorants were extracted from the fruit peel of *S. samarangense*, *S. malaccense*, *D. philippinensis*, and *D. discolor*, as well as the outer bark of *E. deglupta*. Resulting chemical profiles were compared with the hydrophilic dye from leaf decoction of *T. catappa*.

**Figure 2 molecules-26-06645-f002:**
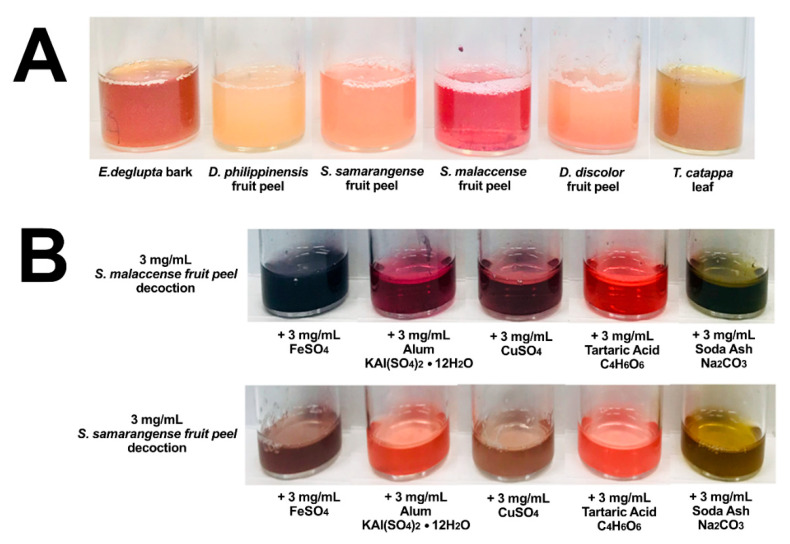
Extracted colorants from biowaste materials. Decoctions from five waste materials and the popular dye source *T. catappa* are displayed in (**A**). In addition, the significant effect of mordants on color is demonstrated in terms of *Syzygium* dyes in (**B**). Additional images from other dye samples can be found in Appendix A.

**Figure 3 molecules-26-06645-f003:**
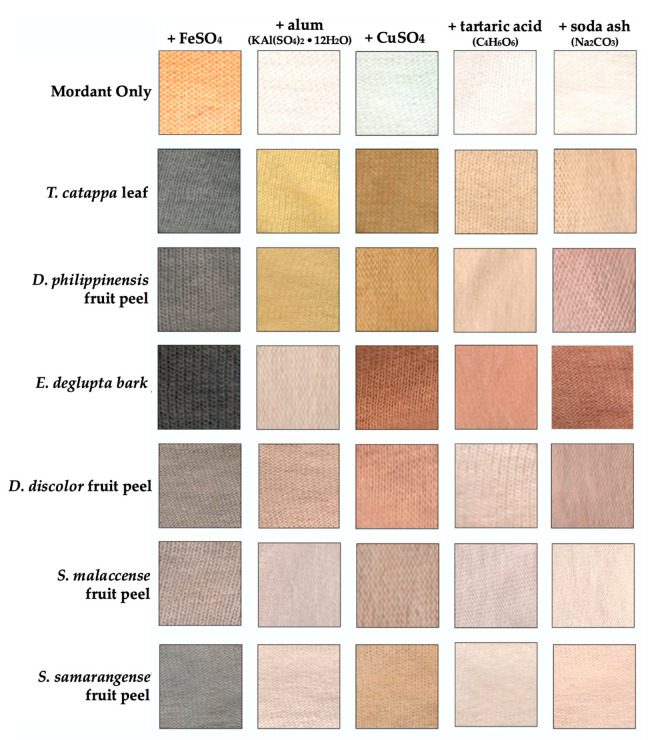
Color palette from the biowaste dyes applied in cotton using different mordants. Comparison of color is through the measured hue (color component), value (lightness), and saturation (intensity). Lighter color (low %V value) is obtained in the presence of FeSO_4_, especially for *E. deglupta* and *T. catappa* (both black). *D. philippinensis* extract mordanted with alum also produced yellow comparable to the well-studied *T. catappa*. Interestingly, red-brown coloration was uniquely observed in *E. deglupta* with CuSO_4_ or alkali.

**Figure 4 molecules-26-06645-f004:**
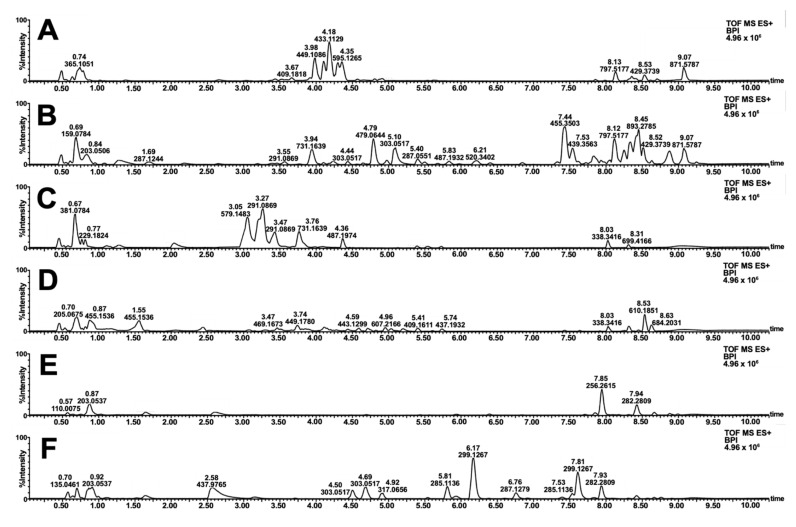
Base peak ion (BPI) chromatogram of extracted natural dyes. Full-scan profiles in the positive mode are shown for decoctions of *T. catappa* leaf (**A**) *D. philippinensis* fruit peel (**B**), *E. deglupta* bark (**C**), *D. discolor* (**D**), *S. malaccense* (**E**), and *S. samarangense* (**F**) fruit peels.

**Figure 5 molecules-26-06645-f005:**
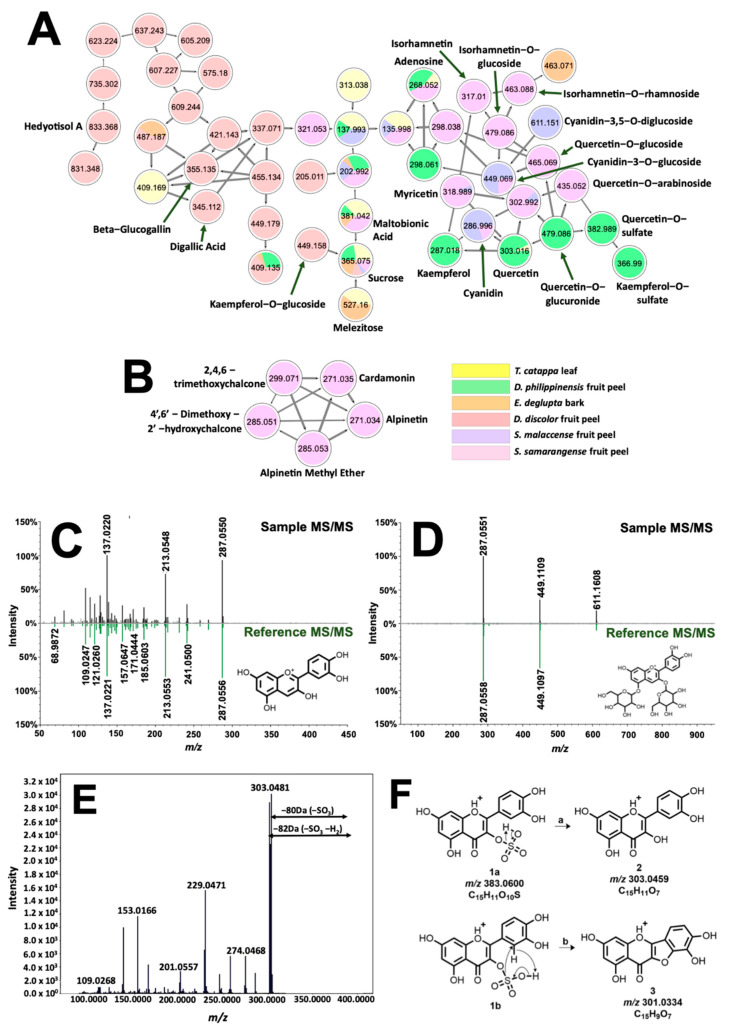
Molecular networking and identification of phenolic compounds from the extracted biocolorants. Structural relationship of flavonoids, sugars, and glycosylated and galloylated compounds are shown in (**A**). Different fragmentation schemes for chalcones in *S. samarangense* led to their exclusive clustering in (**B**). Tail-to-tail alignments for cyanidin and cyanidin-3,5-O-diglucoside are presented in (**C**) and (**D**), respectively. Meanwhile, MS/MS spectra associated with quercetin-3-O-sulfate are presented in (**E**). Fragmentation analysis (**F**) suggested sulfate substitution on the hydroxyl at C-3 of quercetin.

**Figure 6 molecules-26-06645-f006:**
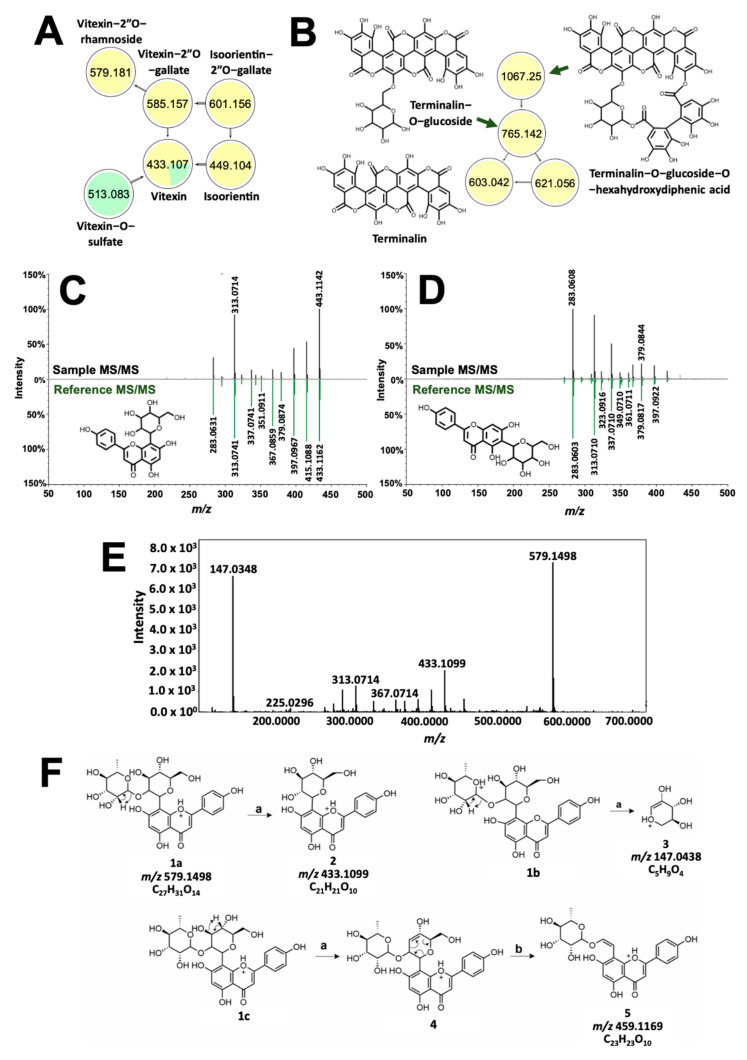
Structural analysis of dye molecules common to *T. catappa* and *D. philippinensis.* Molecular network of flavotannins and ellagitannins are depicted in (**A**) and (B), respectively. GNPS also differentiated positional isomers vitexin (**C**) and isovitexin (**D**). Meanwhile, tandem MS spectrum of *m/z* 579.1496 showed huge similarity with that of vitexin (**E**). Fragmentation analysis suggest the presence of rhamnosyl substituent at C2″ of vitexin (**F**). For the reactions, (**a**) is the remote H-rearrangement, and (**b**) is the RDA reaction.

**Figure 7 molecules-26-06645-f007:**
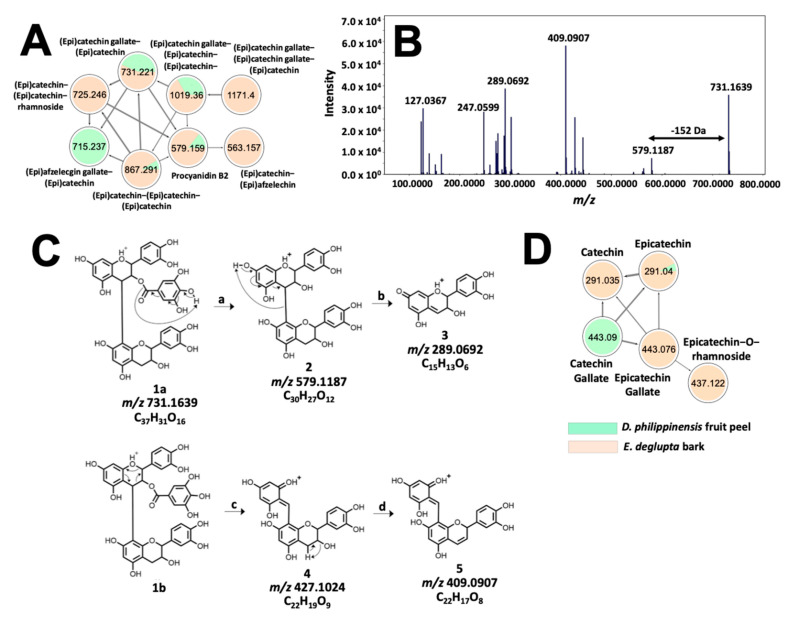
Molecular networking and fragmentation analysis of proanthocyanidins common to *D. philippinensis* and *E. deglupta*. Clustering of complex tannins is shown in (**A**); tandem MS spectrum of *m/z* 731.1667 is presented in (**B**) and served as the basis for fragmentation analysis that characterized the precursor ion as epicatechin-O-gallate-epicatechin (**C**). On the other hand, a smaller network of flavan-3-ol and their galloylated and glucosylated analogs is described in (**D**).

**Figure 8 molecules-26-06645-f008:**
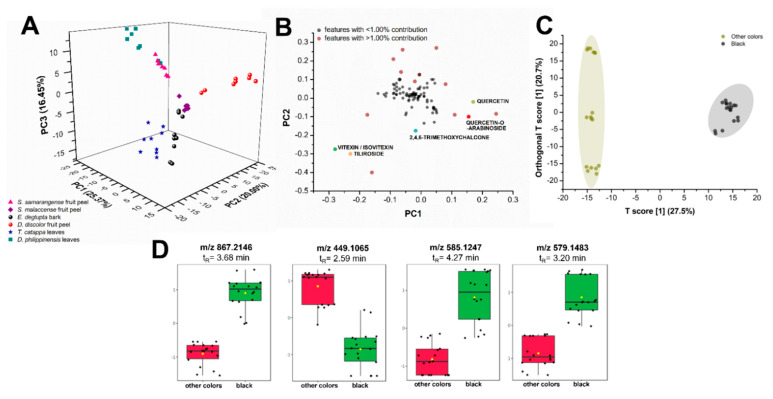
Comparative statistical analyses of metabolite profiles of plant biowaste sources. Unsupervised principal component analysis (PCA) (**A**) with its log-transformed loadings plot (**B**) and supervised OPLS-DA (**C**) show distinct separation between samples according to metabolite constituents and their produced color upon reaction with FeSO_4_ mordant. Putatively identified flavonoids (colored data points) were among the heavily contributing features (**B**) to the dispersion of samples in the PCA scores plot. Samples that produce the black color such as extracts from *E. deglupta* bark, *D. philippinensis* peel, and *T. catappa* leaves show the upregulation of some polyphenols (**D**) such as *m/z* 867.2146 (epicatechin), *m/z* 579.1483 (procyanidin B2), and *m/z* 585.1257 (vitexin-2”-O-gallate).

**Table 1 molecules-26-06645-t001:** Highly significant and variable chemical features determined by XCMS Online.

Compound	Mass (*m/z*)	*p*-Value ^1^	Contribution to Variance (%)	Source ^2^
Vitexin/Isovitexin	433.1	8.81 × 10^−8^	5.12	*T. catappa*, *D. philippinensis*
2,4,6-trimethoxychalcone	299.1	2.89 × 10^−8^	3.09	*S. samarangense*
Quercetin	303.1	6.05 × 10^−5^	4.25	*S. samarangense*
Quercetin-O-arabinoside	435.1	7.18 × 10^−9^	1.76	*S. samarangense*
Tiliroside	595.1	3.09 × 10^−10^	4.03	*D. philippinensis*

^1^ Highly significant features only (*p*-value < 0.0001). ^2^ Sources where the compound is detected to be most abundant.

## Data Availability

The data presented in the study is detailed on the Supplementary Information. Additional information are available on request from the corresponding author.

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
