# Peer review of "Sustainable Hues: Exploring the Molecular Palette of Biowaste Dyes through LC-MS Metabolomics"

_molecules, 2021, doi:10.3390/molecules26216645_

Round 1

Reviewer 1 Report

Introduction:

1- It is suggested that Figure 1 be moved to Results section. This is a good form of review article. Please see article https://dx.doi.org/10.30509/pccc.2020.166673.1074 and https://dx.doi.org/10.30509/pccc.2021.166734.1090. It is suggested that a paragraph containing an overview of research in this area be added to the introduction.

Results:

  1. Please add more details about the dyeing process.

3- If possible, add UV-Vis spectra to the article. Has the effect of using different solvents been investigated?

4- Please explain more about the phrase "For lipids, further studies are necessary to verify its effects on the dye properties".

Materials and Methods:

5- What is the reason for choosing acetonitrile solvent?

Author Response

Dear Reviewer:

Please find attached our point-by-point response to your comments and suggestions.

Comment 1: The most important strength of this manuscript is the use of renewable and natural compounds. These materials are low cost and widely available. On the other hand, they do not have environmental problems and toxicity. The main weakness of the article is the lack of sufficient explanation about the dyeing process. Since the main purpose of the manuscript is to create a new hue, the dyeing process and the study of fastness properties should be considered in full detail. In addition, the obtained result, especially the fastnessy results, should be compared with the published papers. If the authors can study the effect of the presence of mordant, it will be very helpful in strengthening the results of the manuscript. It is best to summarize the fastness properties in one or more tables.

Response 1: We understand that the dyeing process is complex and consists of several steps that begin from extraction of pigments to adsorption and diffusion of colors into the fibers. We appreciate this suggestion, and we fully acknowledge the limitations of our study. However, we would like to emphasize that the focus of our work described in this paper is more on the application of metabolomics methods in dye-related analysis and preparation. 

The study of dye properties such as fastness, and the in-depth exploration of dye-mordant interactions are outside the scope of our methods and by extension, this paper.

Our expertise on mass spectrometry-based metabolomics allowed us to contribute to the information gap on the small molecule diversities of dye sources, such as in biowaste plant materials. Unfortunately, we cannot extend our MS-based methods to investigate dye-mordant interactions as these experiments would be restricted by compatibility to an electrospray ionization source. 

Our approach is a proof of the feasibility of utilizing mass spectrometry in dye analysis to provide insights on color that may not be immediately perceived from the source, and extract at the beginning of the dye process. Moreover, our method could be expanded to accommodate quality control studies of common dye sources, in addition to alternative biowaste materials.

On a positive note, we are collaborating with the Philippine Textile Research Institute as a result of our work with natural dyes. In this collaboration, we will be able to explore the fastness of the biowaste natural dyes we have investigated, expand it to other textiles available at the institute, and publish these results soon.

Comment 2: It is suggested that Figure 1 be moved to the results section. This is a good form  of review article.

Response 2: Revision was made based on the suggestion. Additional paragraph was added in the introduction to accommodate more in-depth discussion about the prospects of natural dyes research. We also cited the suggested papers and also looked at additional works of the authors as reference.

Comment 3: Please add more details about the dyeing process.

Response 3: Additional information on the dyeing process was provided on the methodology section (see page 16, 4.6) and an overview was also provided in the results section. Some of the concerns were also tackled in comment 1.

Comment 4: If possible, add UV-Vis spectra to the article. Has the effect of using different solvents been investigated?

Response 4: For the study, only water was used as extraction solvent. The method was heavily based on the traditional extraction method by our community collaborators. Since the research must directly benefit them, we refrained from using organic solvents that are inaccessible to the community (due to government regulation and potential hazards). We have indicated this as additional information in the revised version of the paper.

As for the UV-Vis analysis, we were able to look at UV-Vis spectra of only T. catappa and D. philippinensis in the presence of FeSO4, alum, and CuSO4. We presented our methodology and used the UV-Vis results as supporting details in the Discussion. Being utilized this way, we prefer that it is presented in the Supplementary Info. 

Comment 5: Please explain more about the phrase “For lipids, further studies are necessary to verify its effects on the dye properties”.

Response 5: Potential advantages and disadvantages of liposomes on dyeing are now incorporated in the Discussion section.

Comment 6: What is the reason for choosing acetonitrile solvent?

Response 6: Acetonitrile is an easily accessible and commonly used solvent in reverse-phase (RP) liquid chromatography. Majority of reverse-phase LC-MS analysis use acetonitrile as mobile phase due to its accessibility, and high compatibility with the electrospray ionization source. Our group has already previously demonstrated the use of acetonitrile in LC-MS analysis of indigo (Molino & Junio, RCMS, 2020). This has been included in the manuscript.

Thank you very much.

Reviewer 2 Report

The article "Sustainable Hues: Explorig the molecular palette of biowaste dyes through LC-MS metabolomics" addresses an urgent concern for the sustainability of the textile industry.

I just have a few comments:

Page 2, figure 1 - Please centre the image with the text, as figure 2. Also, instead of adding what I believe to be a coin, please add a ruler with a scale. It's much more efficient.

Page 2 - please extend what you named as a traditional method o decoction.

Page 2 - please also provide the structure for alum and tartaric acid, as you have for the rest of the mordants

Page 5, Figure 4 - the same comment as with figure 1. Also, the y axes are not very redable. 

Page 7, Figure 5 - The A and B parts are not readable at all. If someone would print, it would not be possible to read anything. On the computer, you really need to zoom in. Please consider another format.

Page 10, Figure 7 - Part A and D are a bit more visible than Figure 5, but not much. It's better to change the figure, or separate it.

Page 16 - you didn't use a colorimeter to analyse your samples?

Discussion: I would like to see a better discussion on the importance of this study, and possible studies on how to overcome the fading of such dyes. Afterall, the authors did use compounds highly susceptible to light, such as flavonoids.

Author Response

Dear Reviewer:

Please find attached our point-by-point response to your comments and suggestions.

Comment 1: Page 2, figure 1 - Please centre the image with the text, as figure 2. Also, instead of adding what I believe to be a coin, please add a ruler with a scale. It's much more efficient.

Response 1: Image was formatted as suggested.

Comment 2:  Please extend what you named as a traditional method of decoction.

Response 2: Additional details are provided in the Discussion   section of the paper. A big part of our methodology, especially with the dyeing process, is influenced by the traditional approach by our community partners.  Justification for the method is the availability and relative safety of water compared to organic solvents.

Comment 3:  Please also provide the structure for alum and tartaric acid, as you have for the rest of the mordants.

Response 3: Chemical formulas of tartaric acid and alum were provided as suggested.

Comment 4: Page 5, Figure 4 - the same comment as with Figure 1. Also, the y axes are not very readable

Response 4: Image was centered, and axes labels were made more readable.

Comments 5: Page 7, Figure 5 - The A and B parts are not readable at all. If someone would print, it would not be possible to read anything. On the computer, you really need to zoom in. Please consider another format.

Response 5: High-resolution format of the image is included in the revised manuscript.

Comment 6:  Page 10, Figure 7 - Part A and D are a bit more visible than Figure 5, but not much. It's better to change the figure or separate it.

Response 6: High-resolution format of the image is included in the revised manuscript.

Comment 7:  Page 16 - you didn't use a colorimeter to analyze your samples?

Response 7: We do not have a hand-held colorimeter device available in our laboratory. Despite this, we still made attempts to assess color through handheld colorimeter apps using a smartphone camera. The results we obtained were comparable to the color values we measured from scanned images analyzed using the Digital Color Meter Tool available in the Apple platform

Comment 8: Discussion: I would like to see a better discussion on the importance of this study, and possible studies on how to overcome the fading of such dyes. Afterall, the authors did use compounds highly susceptible to light, such as flavonoids.

Response 8: Discussion section was edited to mention literature that provides suggestions on how to improve lightfastness of flavonoid-based dyes.

Thank you very much.
